# Analytical Listening and Aesthetic Experience in Music Criticism

**Srđan Teparić** 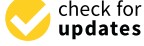

Department of Music Theory, Faculty of Music, University of Arts, 11000 Belgrade, Serbia; teparic@fmu.bg.ac.rs

**Abstract:** In this article, I discuss the methodological and contextual aspects of writing music criticism, drawing cues from applied musicology and autoethnography. The challenge for any music critic is the question of the relationship between objective and subjective approaches. I analyze the relationship between analytical listening and aesthetic experience, using the examples of two music reviews of Ivo Pogorelić's piano recitals that I wrote. The interpretations of this pianist are suitable for the analysis precisely because he is commonly seen as an unconventional, even controversial pianist, and his interpretations of romantic music are often regarded as examples of anti-academicism and even deconstruction of pianistic canons accumulated during the 20th century. Against that term, I will talk about liberation, which is perhaps a more suitable label for Pogorelić's modernist approach to performance.

**Keywords:** music criticism; analytical listening; aesthetic experience; deconstruction of the canon; liberation; perception; evaluation; autoethnography



## 1. Introduction

Applied musicology is an emerging discipline that aims to encompass numerous facets of non-academic work that can contribute to the benefit of society in various ways. As recently discussed by Ivana Medić in her "manifesto", applied musicology can be understood as a "hybrid" discipline situated at the cross point of science, culture, and art, which should "strive to preserve and promote musical heritage both in local and international contexts through performing practices, safeguarding initiatives, curatorial activities, creation of digital archives, concert programming with guided listening, and other forms of public dissemination of research results" (Medić 2022, pp. 89–90). Taking cues from ethnomusicology, Medić denotes six (non-exclusive) areas of applied musicological work: (I) media and new technologies; (II) organization of events; (III) artistic–theoretical work; (IV) archival and curatorial work; (V) cultural policies and activism; and (VI) educational activities (Medić 2022, pp. 90–91). She includes music criticism in her third category alongside other types of writing on music publicly available to all.

In this article, I focus on music criticism as one of the most important segments of applied musicological and music theoretical work. I analyze the relationship between analytical listening and aesthetic experience, using the examples of two music reviews of Ivo Pogorelić's piano recitals.[1] The interpretations of this pianist are suitable for the analysis precisely because he is commonly seen as an unconventional, even controversial performer, and his interpretations of romantic music are often regarded as examples of anti-academicism and even deconstruction of pianistic canons accumulated during the 20th century. Against that term, I will talk about "liberation" which is perhaps a more appropriate label for Pogorelić's modernist approach to piano performance.[2] These approaches not only represent a challenge but also require solidly based arguments that would be exclusively related to a deeper analytical interpretation, as well as the skill of interpreting gestures and embedded meanings in this artist's performances. By analyzing music criticism concerning these two approaches (deconstruction vs. liberation), the following scheme will be used: perception—objective/subjective—comparison—evaluation. The aforementioned points will be explained through examples from the two reviews

of Pogorelić's recitals, guiding us to a possible methodology for defining and outlining features of this specific musical–literary genre.

My personal experience with music criticism and other domains of applied musicology started with practice and ended with theory. Such an approach points to autoethnography, which could be accepted as entirely legitimate within the branch of applied musicology based on an artistic–theoretical basis: "Autoethnography refers to writing about the personal and its relationship to culture. It is an autobiographical genre of writing and research that displays multiple layers of consciousness" (Ellis 2004, p. 37). The legitimacy of this approach, even without explicit reference to the current zeitgeist, lies in the fact that the sensibility of the contemporary era is inextricably linked to personal experience, which is validly acknowledged at the moment when it is incorporated into a well-developed methodology. Remarks and observations related to the contemporary way of interpreting music serve as a framework in which the author's thoughts about the world at large are interwoven. Although I always address my listeners and readers in the first person plural (which is customary in Central and Eastern European academic writing), what I attempt to express points to my current reflections on the contemporary moment, which is necessary for any kind of autoethnography. The subject of temporal events—musical interpretation—begins and ends within the lived-in present. At the moment of writing about somebody's interpretation of a musical piece, the present appears the closest to the past, and for this reason, it would be almost impossible to extract the dose of subjectivity that is necessary in the given circumstances. Therefore, following Ellis, I could classify this autoethnographic approach as qualitative: "The label refers to a variety of research techniques and procedures associated with the goal of trying to understand the complexities of the social world in which we live and how we go about thinking, acting, and making meaning in our lives" (Ellis 2004, p. 25). In my case, the practical use of acquired knowledge and listening experience was fortunately connected to my work in the media.

If I wanted to summarize my own experience, I could highlight three key reasons why, in the quest to find a methodology for writing music criticism, the combination of autobiography and ethnography could emerge as the only logical one. The first reason for reaching for the autoethnographic method is that music criticism incorporates a part of personal experience that is usually bypassed in traditional scientific methods, as pointed out by Wall: "Traditional scientific approaches, still very much at play today, require researchers to minimize their selves, viewing themselves as a contaminant and attempting to transcend and deny it" (Wall 2006, p. 2).

The notion of music criticism, as I understand it, combines personal psychological impressions with signals of what is happening within society itself at any given historical moment. Starting from subjective projections, this approach leads to social connotations or what is often called zeitgeist, which may be the key reason for reaching for autoethnography, as defined by Wiley: "With roots in anthropology, autoethnography is a type of qualitative social science research comprising autobiographical narrative integrated with interpretation and analysis of the wider sociocultural context in which the individual operates, in order to reach an enhanced understanding of the relationship between them" (Wiley 2022, pp. 76–77).

The third reason for reaching for autoethnography is that it points to a dialogic relationship between personal reflection about the world at large and social connotations. Such insights are open to an endless dialogue that could be transmitted through time, and which, on the other hand, speak precisely about the hermeneutics of the time in which we live. Every musical interpretation is a self-reflexive "story" that, when presented to the audience, becomes the subject of interpretations that can move in all possible directions. In other words, they also become "stories". The goal of this article is precisely to open space for further interpretations and discussions through the combination of the personal and the general or, as highlighted in an autoethnographic study intended for piano teachers: "Our story creates a virtual world for the reader and has the capacity to evoke interest and open up a serious discussion" (Gouzouasis and Ihnatovych 2016, p. 24).

When I first started writing music reviews, I understood many of the above concepts only intuitively. Namely, I have worked as a music critic for Radio Belgrade and other outlets for more than twenty years, yet I never received any official training on how to write music critiques and reviews (except for informal feedback from my colleagues at Radio Belgrade). This is because, until very recently, there were no available courses on music criticism at the Faculty of Music in Belgrade, where I studied (nor at any other academic institution in Serbia). Only in the academic year 2020–2021 did the Faculty of Music introduce a new 1-year MA program (60 ESPB): "Primenjena istraživanja muzike" (Applied Research on Music). This study program aims to prepare students to understand phenomena related to the role of music in the modern environment, and to adopt and apply knowledge about the entrepreneurial approach to music, socially engaged artistic/musical practices, and the possibilities of utilizing contemporary digital technologies. I seized the opportunity to prepare and teach a course—Music Criticism and Contemporary Media (MGSM1)—together with my colleagues Marija Masnikosa and Radoš Mitrović. The idea is that students should acquire applicable knowledge in the field of music criticism and, in particular, how to write about music for all contemporary media outlets, including online magazines, blogs, vlogs, and social media.

When it comes to music criticism, the biggest "problem" is the relationship between objective and subjective approaches, because such relations are difficult to measure. No matter how objective a competent music critic might be when evaluating the technical aspects of a performance, there is also a necessary dose of subjectivity that would relate to the emotional experience provoked by the interpretation. Therefore, a valid assessment requires analytical listening, which involves the skill of combining subjective and objective approaches. The aesthetic experience of analytical listening has certain levels of insight that necessarily lead to the last stage of assessing an interpretation, namely, its evaluation. I opted to use illustrative examples of Ivo Pogorelić's piano recitals for considering the methodology of writing reviews, because Pogorelić's performances have always been accompanied by controversies, and this mainly refers to constant discussions about his deconstruction of pianistic canons accumulated throughout the 19th and 20th centuries. It is precisely the judgments related to these kinds of concerns that most influence the evaluation. Although I am often not aware of the stages that follow analytical listening and the act of evaluation, I will now attempt to break down the process of writing music critiques, starting with analytical listening.

Reviews that I write for Radio Belgrade are specific in that they are limited by the time that the critic has at their disposal (usually between three and six minutes of airtime); hence the radio critic must "trim the fat" and focus only on the most important and memorable facets of someone's interpretation. The two music reviews I wrote for Radio Belgrade 2 after Ivo Pogorelić's recitals held in the Great Hall of the Kolarac Endowment in Belgrade on 18 December 2017 and 7 November 2022 will exemplify my methodology.

## 2. The Context of Pogorelić's 2017 Belgrade Performance

Born in 1958 in Belgrade (then the capital city of the Socialist Federal Republic of Yugoslavia, today of the Republic of Serbia) to a Croatian father and Serbian mother, but nowadays a citizen of Switzerland (where he resides) and Croatia, throughout his career Ivo Pogorelić has been celebrated and criticized in equal measure. Starting from his sensational elimination in the third round of the Tenth International Chopin Piano Competition in Warsaw in 1980, which prompted juror Martha Argerich to resign from the jury in protest and to call Pogorelić a "genius", his highly original and sometimes even eccentric performances have provoked controversies and fierce debates. Nevertheless, he had a very successful career, releasing 14 best-selling albums for Deutsche Grammophon between 1981 and 1995 and achieving true celebrity status in his native Yugoslavia. For example, composer and Fellow of the Serbian Academy of Sciences and Arts Mihailo Vukdragović (1900–1986) wrote in an aside to Pogorelić's interview with the weekly magazine *NIN* on 29 December 1980: "I have worked in music for half a century, both as a composer and as a performer. I

have listened to all the virtuosos of the world, but I have never heard anything like this. And, if it is true that the great Horowitz said after listening to Pogorelić's playing that he can now die peacefully, he is right: he really can die" (Đurović 1980, p. 30). As we can deduce from this exaggerated statement, almost immediately after the Warsaw scandal, Pogorelić acquired the halo of a cult personality in his country. Yet, even in Yugoslavia, there were different opinions on his unconventional interpretations. In stark contrast to Vukdragović, renowned musicologist, composer, and aesthetician Dragutin Gostuški publicly gave Pogorelić some unsolicited but well-meaning advice: "The worst and the best that can be said about him now is that he is (although he might want to be) neither a fraud nor a genius, but just an excellent musician who tries to impress us by constantly defying the general musical opinion" (Gostuški 1980, p. 31).

There was a 28-year gap between Ivo Pogorelić's 2017 recital and his previous Belgrade performance. Namely, Pogorelić performed his final concert in his native city on 28 March 1989, i.e., before the dissolution of the SFR Yugoslavia and the ensuing wars, which pushed many artists into exile and, even worse, forced them to take sides. This situation was especially challenging for artists of mixed national and religious backgrounds. The Belgrade audience has always been partial to their favorite performers, such as acclaimed violinists Stefan Milenković (b. 1977) and Nemanja Radulović (b. 1985), who have similarly achieved cult status (and also spent many decades living abroad—Milenković in the USA, Radulović in France). However, these artists continued to perform regularly in Belgrade (and the rest of Serbia), whereas Pogorelić had not played in his native city for almost three decades, which reinforced the myth of a unique genius in absentia. The reasons for such a prolonged absence of Pogorelić from Belgrade concert life were the devastating Yugoslav wars and their aftermath, but also the artist's own prolonged hiatus after the death of his first wife Alisa Kezheradze in 1996, pushing him into a profound personal and artistic crisis. Thus, the announcement of Pogorelić's return to Belgrade after a 28-year-long self-imposed exile caused great excitement and even euphoria among the concertgoing audience.

## 3. On Methodology, or: Four General Points about Writing Music Critiques

I began my review of Pogorelić's 2017 "homecoming" Belgrade recital by liberating myself from any preconceptions related to the context of the pianist's performance and the surrounding expectations; thus, I wrote: "Before Ivo Pogorelić's actual recital, the author of this review tried his best abstain from reading any information regarding his arrival in Belgrade after so many years. There was a desire to experience the interpretation as directly as possible, and the text that follows should represent a more-or-less objective analysis of the technical aspects of the performance, but still, from an inevitably subjective viewpoint" (Teparić 2017). This short excerpt from an actual review leads us to four points that should be considered and classified methodologically:

1.  Critical discourse necessitates the use of hermeneutic interpretation (the branch of knowledge that deals with the interpretation of literary texts); according to Gadamer, "To understand the statement of a text, starting from the concrete situation in which that statement was created, is a pure hermeneutic requirement" (Gadamer 1978, p. 369).

2.  The critic should be free of all prejudices, although this is not entirely possible; some will remain and, for that reason, it is recommended not to read the concert program until the first notes are played; and, if the performer is not well-known, the critic should avoid reading the biography or anything that might create certain expectations or prejudices.[3] In this way, a free space opens up for immersion in analytical listening and for the meeting of intellectual and emotional experiences, which ultimately results in an aesthetic evaluation.

3.  Analytical listening stems from previously acquired knowledge, i.e., it implies exceptional competence of the critic both in the realm of music-technical disciplines such as harmony, form, and counterpoint, and a broad knowledge of aesthetic and stylistic premises related to compositions that are performed, and to the performance itself.

4. The combination of intellectual and emotional competencies should lead to an objective analysis of the technical aspects of performance, albeit with necessary subjectivity, especially when it comes to the evaluation of the interpretation. Subjectivity is inevitable because analytical listening necessarily involves the activation of emotions.

Now let us return to the review itself in order to confirm or counteract these points:

> Pogorelić's interpretation of Mozart's Fantasia in C minor K 475 pointed to the deconstruction of some of the canons related not so much to the performance of Mozart himself, but to the connection of tones in a melodic sequence in the narrower sense, and harmonic phrases in the overall statement in a wider sense. In the performance of this piece, the pianist seemed to devote a short time, say the smallest fraction of a second, to the consideration of the individual properties of each tone with every touch of his fingers on the keyboard (Teparić 2017).

As far as the interpretation of Mozart is concerned, the remark about deconstruction comes to the fore. As aptly observed by Norris, deconstruction is a critical term that could be associated with some kind of crisis concerning the context of the term being discussed: "Deconstruction is a constant reminder of the etymological connection between 'crisis' and 'criticism'. It makes manifest the fact that any radical shift of interpretive thought must always come up against the limits of seeming absurdity" (Norris 2002, p. xii). Thus, understood in this way, deconstruction leads us to the annulment of meaningfulness. In a semiotic sense, it points to the liberation of the sign from its expressive function and its reduction to a mere signifier, an act worthy of earlier structuralist criticism.[4] As regards Pogorelić, his deconstruction is simultaneously critical and creative, and at that point, the discourse of criticism coincides with the specific interpretation of the pianist.

Derrida's well-known challenge to speech and logocentrism could be comparable to Pogorelić's challenge to the baggage of old interpretive formulas accumulated over the centuries.[5] If we were to reach for this term in connection with Pogorelić, deconstruction would refer to his (overly) analytical playing. The specifics of his approach lead to the conscious division of musical expression into smaller parts, then to their recomposition, whilst assigning them new meanings. In the act of music performance, the listener gets a clear impression of that process because the performer himself deliberately separates the phrases in such a way that the seams of his musical fabric appear as not fully attached. On the other hand, the critique should review the outcome of Pogorelić's "deconstruction", which actually looks more like a release from the canonical constraints and a striving towards (re)discovering the long-lost meaning: "This is what makes Pogorelić unique and justifies his alleged aloofness. In essence, there is no aloofness—instead, Pogorelić's interpretations are simply liberated from the burden of descriptiveness and narrativity associated with romantic pianism. Its linguistic signs operate within a system that is far removed from its everyday conventional use within the long-petrified pianistic canons" (Teparić 2017). The same is observed with respect to Pogorelić's interpretation of Beethoven's Sonata in F minor Op. 57 no. 23 (*Appassionata*): "In a sound universe freed from the literal description, the drama expressed by musical means really becomes dramatic, the chorale from the beginning of the second movement turns into an overtone-filled musical-liturgical rite, while lyrical melodies are purified to the level of light 'ear-tickling' movements of the pre-classical period" (Teparić 2017).

## 4. Four Methodological Elements

Regarding Ivo Pogorelić's performances of romantic compositions such as Jean Sibelius's *Valse Triste* and Sonata No. 2 in B flat minor by Sergei Rachmaninov, this is where his "deconstruction" becomes most apparent. Looking at the aesthetic experience of analytical listening translated into music criticism, one cannot talk about gradual steps of perception, because there is no strict hierarchy of methodological tools. In an article dedicated to music reviews broadcast on Radio Belgrade 2, Sanela Nikolić presents a five-level model of aesthetic experience: (1) perceptual analysis, (2) implicit memory integration, (3) explicit

classification, (4) cognitive mastery and (5) evaluation ([Nikolić 2020](), p. 197).[6] If we apply Nikolić's model to our review of Pogorelić's 2017 piano recital, the perceptual analysis would concern the perception of performing aspects; implicit memory integration would concern comparisons between this particular recital and other performances of the same pieces; the third level, explicit classification, implies the integration of the previous two levels and describes what happens during the interpretation; cognitive mastery is the level of expertise, that is, musical analysis; and the fifth level refers to the final evaluation of the performance. If we wanted to summarize these levels into practical methodological considerations and instructions for writing music criticism, one should start with perception and analytical consideration of structural, formal and/or technical aspects of interpretation, which, in the review itself, do not have to be devoid of aesthetic or psychological aspects. For example, this is what I wrote about Pogorelić's interpretation of Rachmaninov: "Complex passages, dynamic peaks, virtuosity as such—all these wonderful stunts seemed as logical units of a disassembled whole, carefully reassembled into new formal patterns" ([Teparić 2017]).

The second tool is an attempt at objective perception with the necessary inclusion of the subjective experience, because the critical discourse does not exclude the emotional aspect, as exemplified by this excerpt: "The performance of Sibelius's *Valse Triste* seemed the most extreme, and it appeared that the pianist went beyond the process of recreating the score, approaching the point where he eclipsed the composer. Such impressions are always extremely subjective because they arise from the inexpressible language of emotions; to complement this point of view, one could legitimately state that Pogorelić returned Sibelius to himself, as he had done previously when he so confidently purified Brahms's *Intermezzos* from all layers of false emotionality" (ibid.).

The third aspect would be a direct comparison with previous experiences, which requires the experience and competence of the critic: "For decades, Rachmaninov's music has been vulgarised to the realm of the banalest effects, which are euphemistically dubbed 'sumptuous musicality'. However, sumptuous musicality carries with it a huge responsibility, because it should develop both towards the surface and the depths of the inner being of the performer" (ibid.).

Evaluation is the ultimate instance of any criticism, as it offers a possible final aesthetic judgment. It requires the synthesis of all the facts and contexts presented previously. In the next example, I characterized Pogorelić differently from Vukdragović, who considered him an unadulterated genius, or Gostuški, who assessed him as an excellent musician who simply wanted to impress and stun the audience. The contexts presented in the following excerpt are linguistic, social, political, philosophical, aesthetic, and psychological.

> Finding their unique language is the dream of every artist, and performers have the unenviable task that, in addition to transmitting what has already been created, they must also upgrade and complement it with their own creation. Pogorelić is one of the few who possesses the power and integrity to reach the points of interpretation where the performer's and the composer's contributions merge into an original creation. First and foremost a creator, and then an interpreter, Pogorelić acts as an avant-garde revolutionary who fights against the consumerist burdens with which musical performers of the most diverse genres try to please the audience. The individual versus the mass is an archetype that has been repeated for centuries, and Pogorelić has taken this heavy burden upon himself. Therefore, he is not a typical artist of this time, just as he would not be typical in any other period. His 'hero' qualities are timeless, and it seems that he has managed to capture the universal language of the human spirit through his playing. And, speaking of spirit, we are here in the realm of romanticism, but not the sugary, saturated, ear-pleasing, salon-type of romanticism, but the one in which music was considered the language closest to the true nature of the inner depths of being (ibid.).

Hence, our methodological scheme can be summed up as such: the analytical listening of a competent critic activates the mechanisms of acquired knowledge and previous aesthetic experiences, in order to connect separate elements of the interpretation into a meaningful whole, the context of which is always unique.

## 5. Five Years Later

Ivo Pogorelić's Belgrade recital on 7 October 2022 was again sold out.[7] Apart from the fact that the pianist lost nothing of his cult status, it is remarkable that he presented a program in which the performing canons were most firmly established. The concert was dedicated to the music of Frédéric Chopin and the following compositions were performed: Polonaise-Fantaisie in A flat major, Op. 61, Piano Sonata No. 3 in B minor, Op. 58, Fantaisie in F minor, Op. 49, Berceuse in D flat major, Op. 57 and Barcarolle in F sharp major, Op. 60.

I have already mentioned that the inclusion of methodological elements is done alternately, so my review of this concert began with an evaluation, as if continuing from where the previous one left off: "Romanticism as a great narrative received one of the most complete revisions in the interpretation of Ivo Pogorelić" (Teparić 2022). We would have to stop at this first sentence, because the question could be raised whether we can put an equal sign between revision and deconstruction, although my preferred term is the "liberation" of Romanticism. This would specifically mean the liberation of romantic musical statements from the deposits of the 20th-century pianistic canons, i.e., from the traces of the affective and overemphasized mannerism, i.e., from everything that Pogorelić himself considers to be an unnecessary addition.[8] "In all that flow of phrases in Chopin's music, there were some obvious seams, but not abrupt breaks, so the formal units melted into one another. Because of this, the listener had the impression that more than deconstructing and reducing Chopin's universe to mere signifiers, Pogorelić is returning to where it always belonged—to the forgotten spiritual world of romantic art as imagined by the artist himself" (Teparić 2022). Five years onward, it seems that the music critic himself managed to get to the essence of Pogorelić's art of interpretation, so the process of analytical listening was carried out even more profoundly than in the previous case. Of course, the interpretations concerning the "seams" within the formal patterns of Chopin's compositions would not be possible without the previous experience of listening to Pogorelić's live performances.

If we were to make a comparison with any other pianistic practice, Pogorelić's search for a specific structure of a piece of music is perceived as a kind of transgression, and this especially concerns the academic approach that Pogorelić negates with his playing:

> This pianist treats the form in such a way that it is experienced as a series of varied statements, within which freedom of interpretation is allowed in the sense that each ensuing phrase, although similar, is nevertheless different from the previous one in some detail. Such a stringing of constantly varied phrases produces a collage-like set of 'patches' that are not devoid of meaning within themselves (Teparić 2022).[9]

Embracing the form in the listener's mind does require competence and the knowledge that traditional formal patterns such as sonata form or tripartite form are relativized. Musicologist and music critic Stefan Cvetković uses the term "modernist transgression" to describe Pogorelić's liberating approach to piano playing:

> Nevertheless, as far as Pogorelić's efforts in the field of tonal expression can be explained by the origins of his mastery stored in the accumulated knowledge of the great tradition from which he emerged, and in which his pianism is perceived as a defense and an extension of continuity, at the same time it serves as an undisputable link to the characteristic modernist transgression, in terms of the aspects of the novelty that he introduced. In that sense, it would not be wrong to claim that the richness of Pogorelić's sound images is the result of an individual approach to the structure of the musical work, in which he finds multiple sign-

posts to the possibilities of interpretation, and which precisely represents a solid modernist character (Cvetković 2022, pp. 259–60).

In Cvetković's view, the transgression is of a modernist nature, because Pogorelić's interpretation of Chopin seems to him anti-romantic, hence modern. On the other hand, my review addresses the idea that romanticism has returned to itself, i.e., that it is the kind of understanding of style that leads to the pianist interpreting the era in the way he thinks it could have sounded at that time. This approach is particularly evident in the so-called "authentic" or "historical" performances of early music, but the same can apply to Pogorelić's interpretations of early romanticism, whereas the striving to return to the "authentic" sound of the past is itself a modernist phenomenon. Writing about Pogorelić's performances necessarily involves comparisons from which, finally, comments emerge on interpretive interventions that have hitherto not been seen in the interpretation of Frédéric Chopin's music. When it comes to lyricism and drama as the main characteristics of Chopin's music, I wrote:

> Pogorelić seems to want to tell his listeners that it is not only the melody that (in contrast to the accompaniment) must indisputably be highlighted in the foreground. Occasionally it appears so, but already in the next segment, beyond all expectations, we can hear an independent accompanying melody or a figured movement of the left hand that complements the melody and is presented in a completely different context than the one that we are used to. This was especially evident in the *Polonaise-Fantaisie* in A Flat Major and the slow movement of Sonata No. 3 in C minor. The latter example was perhaps the most exciting point of the concert because that movement did not seem to be directed (only) forward, but it seemed to spring from the depths, which was expressed through an extremely impressive piano in which the chords and melody seemed to be drawn from the deepest layers of the pianist's being, or perhaps the collective being. Precisely because of this, it seems that Pogorelić presented romanticism to the audience as an archetype, and as we know, romanticism is, first, freedom, and only then—convention. Thus, in Pogorelić's interpretation, even the drama of the Sonata in B minor or the Fantaisie in F minor was not a stormy and annihilating force (Teparić 2022).

This excerpt shows that the hermeneutic pattern, which implies the perception reached through analytical listening, simultaneously implies objective observation—otherwise, such actions would have been characterized as gross violations of the expectations derived from academic stereotypes. I am referring to the cases where the lyrical melody is so emancipated from the accompaniment that it seems sugared, while the fast passage movements are places where the pianist's virtuosity is more noticeable than Chopin himself intended. This, in turn, leads to a questioning of the modernist negative prejudice against the romantic era. Pogorelić's romanticism thus receives new shapes, discovered by the pianist himself, which has led me to claim tentatively:

> In certain chordal sounds, we could hear different aliquots and collages of similar colors spread along the vertical sound image. Perhaps Pogorelić's colorism is one of the key points that bring back a sort of forgotten coloristic world of the original romantic sound to modern pianism (Teparić 2022).

All of the above indicates that the process of analytical listening ultimately involves the objective breakdown of the elements of interpretation and highlights precisely those characteristic elements that are identified by comparison with what could be considered to be expected. The perceptions of the treatment of form, shaping of sound, interpretation of lyricism and drama, impose themselves as objective factors that we could assess in different ways. The subjective assessment of the music critic refers to the fact that the artist is fully capable of introducing the listener to another, fictional world:

Pogorelić must be recognized for his ability to absorb the listener into his world. And his world is special, sublime, and suggestive. Taking responsibility for highlighting just these three points, Pogorelić showed that the freedom of his playing is neither chaotic nor nihilistic—on the contrary, it is creative and, within it, a whole firmly built system that translates Chopin into sensitivity, into a man-sized drama, or a sublime mysticism, is generated. Thus Pogorelić's interpretation of Chopin's music is profoundly humane (Teparić 2022).

With this statement, the hermeneutic circle, which consists of perception, an objective analytical disassembly into parts, their comparison, reassembly of the whole, and its subjective evaluation, is closed again. Just like the performance itself, music criticism is an act of interpretation that additionally translates into words what was "said" on the stage with notes, and by doing so reminds us that every work of art and its interpretation represent a common good that has the potential to be further upgraded and loaded with different meanings.

**Funding:** This research received no external funding.

**Data Availability Statement:** The original contributions presented in the study are included in the article, further inquiries can be directed to the corresponding author.

**Acknowledgments:** I would like to extend my sincere gratitude to Ivana Medić for the invitation to collaborate on this special issue on applied musicology and ethnomusicology, and for her assistance in translating the article. I would also like to thank the two anonymous reviewers for their careful reading of the text and insightful comments.

**Conflicts of Interest:** The author declares no conflicts of interest.

## Notes

[1]  His surname is alternatively spelled Pogorelich, in the international context.

[2]  Unfortunately (and surprisingly), so far not a single book or study has been devoted to the art of interpretation of this outstanding pianist. However, in several of his interviews, Pogorelić himself talked about the modernist approach to interpretation, which will be discussed later.

[3]  Gadamer also deals with this hermeneutical question: "Certainly, it cannot be a general assumption that what we are told in a text completely agrees with our opinion and expectations. On the contrary, what someone tells us, whether in a conversation, in a letter or book, or in some other way, we assume that the opinion expressed there is theirs, not ours, and that it should be taken into account without the need to disseminate it. However, this assumption does not facilitate understanding, but makes it more difficult, since our own preconceptions, which determine our understanding, can remain completely unnoticed. If they motivate a misunderstanding—how can we notice a misunderstanding when facing a text where there is no answer from someone else? How can one protect oneself in advance from misunderstanding the text?" (Gadamer 1978, p. 301).

[4]  For example, de Man writes: "The fallacy of the belief that, in the language of poetry, sign and meaning can coincide, or at least be related to each other in the free and harmonious balance that we call beauty, is said to be a specifically romantic delusion. The unity of appearance (sign) and concept (meaning)—to use the terminology that one finds indeed among the theoreticians of romanticism when they speak of *Schein* and *Idee*—is said to be a romantic myth embodied in the recurrent topos of the 'Beautiful Soul'" (de Man 1983, pp. 12–13).

[5]  In one of his statements, Derrida mentions the "indicative", which, following Husserl, refers to "lifeless" signs and stands in contrast to "expressive" signs that carry meaning. In this sense, Derrida states: "Although there is no expression or meaning without speech, on the other hand not everything in speech is 'expressive'. Although discourse would not be possible without an expressive core, one could almost say that the totality of speech is caught up in an indicative web" (Derrida 1973, p. 31).

[6]  Nikolić draws these five levels from (Leder et al. 2004, pp. 489–508).

[7]  In the meantime, Ivo Pogorelić held another concert for which I did not write a review. It took place on 7 November 2018, also in the Great Hall of the Kolarac Endowment in Belgrade. The program included Adagio in A Minor KV 540 by Wolfgang Amadeus Mozart, Sonata in B Minor by Franz Liszt and Symphonic Etudes Op. 13 by Robert Schumann.

[8]  In his doctoral dissertation, *Contextual and Interpretive Aspects of Piano Practice in the Epoch of Modernity*, Stefan Cvetković deals with precisely this question: "Modernist processes in pianism were seen as expressions of a new rational artistic discourse based on an objectified attitude towards the recording of a work. These efforts were intentionally aimed at freeing musical interpretation from traditional settings based on an overly virtuoso or overly expressive playing, or many practices characterized by an improvisational approach to musical work typical of 19th-century pianism" (Cvetković 2022, p. iii). Cvetković shows that

throughout the 20th century, these stereotypically romantic performing practices were gradually abandoned in favor of more rational approaches.

9     It seems that the source of Pogorelić's "modernism" lies in the interpretation of Ludwig van Beethoven, in whose music the pianist found the key to connecting the past and the present: "Beethoven is the most important figure for the formation of modern pianism. All my life I have searched for this original, primary source. I never liked listening to Beethoven's music on the piano, it was not convincing, even under my fingers. I have always had the impression of a distance, like a filter, between our present epoch and the old one, which is beyond our reach" (Boissard 2013). In a later interview, Pogorelić admitted that Beethoven was the source of his modernism: "Beethoven rewarded me, and some of what I had to seize from him as a professional pianist later returned to me as a bonus, not just for the performances of Beethoven himself, but also for other composers, such as Rachmaninoff. Thus I opened a new window that allowed me to look through it and see the goal, which now seemed a bit farther away than before, and to set new tasks for myself" (Pofuk 2020).

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
