# Peer review of "Analytical Listening and Aesthetic Experience in Music Criticism"

_arts, 2021_

Round 1

Reviewer 1 Report

Comments and Suggestions for Authors

An original and well-presented topic. An unexpected methodological angle, given that autoethnography is more commonly applied to artistic rather than musicological activities.

At first, attributing Pogorelich to "modernist" approach raised a doubt, but during the course of the article the argument was based in a convincing way.

I may have reservations whether in contemporary culture we can avoid any preconception before going to a concert/writing a review - I would rather accept the fact that those preconceptions are an inevitable part of the whole process. But the author has the right to a different opinion.

The article has clear methods, well-presented context and valuable insights. The only thing that may be still improved is slightly elaborating on conclusions.

Author Response

Dear mr/mrs,

First of all, thank you for opinion that my theme is original and well presented. I've highlighted all the revised parts of the text so you can understand what's new. It is true that autoethnography is related to artistic activities. However, music criticism requires a dose of subjectivity, so it seemed logical to me that I used autoethnography in my work. In this regard, in the revised version I explained in more detail why I used this methodology:

(If I wanted to summarize my own experience, I could highlight three key reasons why, in the quest to find a methodology for writing music criticism, the combination of autobiography and ethnography could emerge as the only logical one. The first reason for reaching for the autoethnographic method is that music criticism incorporates a part of personal experience that is usually bypassed in traditional scientific methods, as pointed out by Wall: “Traditional scientific approaches, still very much at play today, require researchers to minimize their selves, viewing themselves as a contaminant and attempting to transcend and deny it” (Wall 2006, 2).

The notion of ​​music criticism, as I understand it, combines personal psychological impressions with signals of what is happening within society itself at any given historical moment. Starting from subjective projections, this approach leads to social connotations or what is often called zeitgeist, which may be the key reason for reaching for autoethnography; as defined by Wiley, “With roots in anthropology, autoethnography is a type of qualitative social science research comprising autobiographical narrative integrated with interpretation and analysis of the wider sociocultural context in which the individual operates, in order to reach an enhanced understanding of the relationship between them” (Wiley 2022, 76−77).

The third reason for reaching for autoethnography is that it points to a dialogic relationship between personal reflection about the world at large and social connotations. Such insights are open to an endless dialogue that could be transmitted through time, and which, on the other hand, speaks precisely about the hermeneutics of the time in which we live. Every musical interpretation is a self-reflexive “story” that, when presented to the audience, becomes the subject of interpretations that can move in all possible directions − in other words, they also become “stories”. The goal of this article is precisely to open space for further interpretations and discussions through the combination of the personal and the general − or, as highlighted in an autoethnographic study intended for piano teachers: “Our story creates a virtual world for the reader and has the capacity to evoke interest and open up a serious discussion” (Gouzouasis & Ihnatovych 2016, 24).

In addition, in footnote nine I used pogorelić's statements in which he explains the roots of his modernism:

(It seems that the source of Pogorelić's “modernism” lies in the interpretation of Ludwig van Beethoven, in whose music the pianist found the key to connecting the past and the present: “Beethoven is the most important figure for the formation of modern pianism. All my life I have searched for this original, primary source. I never liked listening to Beethoven’s music on the piano, it was not convincing, even under my fingers. I have always had the impression of a distance, like a filter, between our present epoch and the old one, which is beyond our reach” (Boissard, 2013). In a later interview, Pogorelić admitted that Beethoven was the source of his modernism: “Beethoven rewarded me, and some of what I had to seize from him as a professional pianist later returned to me as a bonus, not just for the performances of Beethoven himself, but also for other composers, such as Rachmaninoff. Thus I opened a new window that allowed me to look through it and see the goal, which now seemed a bit farther away than before, and to set new tasks for myself” (Pofuk 2020).

Once again, thank you very much for your helpful suggestions. I think because of that, my paper will be even better quality.

Reviewer 2 Report

Comments and Suggestions for Authors

The paper deals with the process of music criticism in a coherent approach based on hermeneutics and presents it in an adequate way illustrating it with a relevant case study.

There are some issues that can be improved to raise its quality and deepness:

- The approach to the concept of autoethnography is a bit forced, serving only to the purpose of self-referencing. It should be more consistent with the actual practice in anthropology and the social sciences. 

- The author's valuable self citations should be completed with a deepen analysis of literature on Pogorelich, reinforcing the state of art, otherwise too narrow.

- For anthropological interest, the words of Pogorelich, himself about his approach to music would be very valuable.

Author Response

Dear Mr/Mrs

First of all, thank you for your suggestions. I've highlighted all the revised parts of the text so you can understand what's new. In the revised version I explained in more detail why I used autoethnography as a methodology:

(If I wanted to summarize my own experience, I could highlight three key reasons why, in the quest to find a methodology for writing music criticism, the combination of autobiography and ethnography could emerge as the only logical one. The first reason for reaching for the autoethnographic method is that music criticism incorporates a part of personal experience that is usually bypassed in traditional scientific methods, as pointed out by Wall: “Traditional scientific approaches, still very much at play today, require researchers to minimize their selves, viewing themselves as a contaminant and attempting to transcend and deny it” (Wall 2006, 2).

The notion of ​​music criticism, as I understand it, combines personal psychological impressions with signals of what is happening within society itself at any given historical moment. Starting from subjective projections, this approach leads to social connotations or what is often called zeitgeist, which may be the key reason for reaching for autoethnography; as defined by Wiley, “With roots in anthropology, autoethnography is a type of qualitative social science research comprising autobiographical narrative integrated with interpretation and analysis of the wider sociocultural context in which the individual operates, in order to reach an enhanced understanding of the relationship between them” (Wiley 2022, 76−77).

The third reason for reaching for autoethnography is that it points to a dialogic relationship between personal reflection about the world at large and social connotations. Such insights are open to an endless dialogue that could be transmitted through time, and which, on the other hand, speaks precisely about the hermeneutics of the time in which we live. Every musical interpretation is a self-reflexive “story” that, when presented to the audience, becomes the subject of interpretations that can move in all possible directions − in other words, they also become “stories”. The goal of this article is precisely to open space for further interpretations and discussions through the combination of the personal and the general − or, as highlighted in an autoethnographic study intended for piano teachers: “Our story creates a virtual world for the reader and has the capacity to evoke interest and open up a serious discussion” (Gouzouasis & Ihnatovych 2016, 24).

Unfortunately, there is no monograph dedicated to Pogorelić. Several of his interviews are available on his website (footnote 2). As you suggested, I used the allegations of the artist himself to prove my theses. They are found in footnote 9 and talk about the interpretation of Beethoven and the artist's attitude towards modernism:

(It seems that the source of Pogorelić's “modernism” lies in the interpretation of Ludwig van Beethoven, in whose music the pianist found the key to connecting the past and the present: “Beethoven is the most important figure for the formation of modern pianism. All my life I have searched for this original, primary source. I never liked listening to Beethoven’s music on the piano, it was not convincing, even under my fingers. I have always had the impression of a distance, like a filter, between our present epoch and the old one, which is beyond our reach” (Boissard, 2013). In a later interview, Pogorelić admitted that Beethoven was the source of his modernism: “Beethoven rewarded me, and some of what I had to seize from him as a professional pianist later returned to me as a bonus, not just for the performances of Beethoven himself, but also for other composers, such as Rachmaninoff. Thus I opened a new window that allowed me to look through it and see the goal, which now seemed a bit farther away than before, and to set new tasks for myself” (Pofuk 2020).

Once again, Thank you very much for the useful insights that have contributed to my work gaining much more quality.
